# EchoX: Towards Mitigating Acoustic-Semantic Gap via Echo Training for Speech-to-Speech LLMs

## Abstract

Speech-to-speech large language models (SLLMs) are attracting increasing attention. Derived from text-based large language models (LLMs), SLLMs often exhibit degradation in knowledge and reasoning capabilities. We hypothesize that this limitation arises because current training paradigms for SLLMs fail to bridge the acoustic-semantic gap in the feature representation space. To address this issue, we propose EchoX, which leverages semantic representations and dynamically generates speech training targets. This approach integrates both acoustic and semantic learning, enabling EchoX to preserve strong reasoning abilities as a speech LLM. Experimental results demonstrate that EchoX, with about six thousand hours of training data, achieves advanced performance on multiple knowledge-based question-answering benchmarks.

## 1 Introduction

GPT-4o (Hurst et al., 2024) demonstrates impressive speech interaction performance, which has spurred the rapid development of speech-to-speech large language models (SLLMs). The mainstream approach to building SLLMs is to first discretize speech into speech tokens and then train speech LLMs (Zhang et al., 2023; Défossez et al., 2024; Chen et al., 2025a) under a token-based training paradigm. Although current SLLMs can be trained on massive amounts of speech data, they still exhibit **intelligence degradation** compared to large text-based models (Chen et al., 2024).

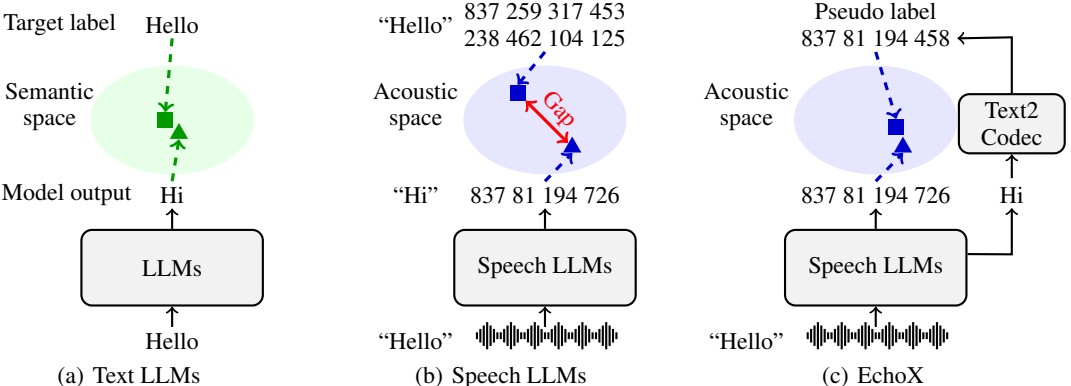

Figure 1: Comparison of training strategies across different models.

Current SLLMs have not yet fully extended the textual intelligence of LLMs into the speech domain, and the underlying reasons for this issue remain underexplored. Beyond the acoustic-semantic conflict in speech

tokens (Gong et al., 2025), we argue that one of the main causes is that SLLMs have not bridged the **acoustic-semantic gap** in the feature representation space. As illustrated in Figure 1(a), the training objective for LLMs emphasizes semantic correctness—predicting a semantically similar token is not heavily penalized. In contrast, SLLMs treat speech tokens as prediction targets, which biases the model toward pronunciation-level accuracy. As a result, even when an SLLM produces a semantically correct response, it may incur severe penalties due to major pronunciation differences, as shown in Figure 1(b).

There are two main paradigms for building SLLMs. The first is interleaved generation (Zeng et al., 2024), which forces the model to jointly consider both acoustics and semantics, but requires a large amount of training data (Chen et al., 2025b). The second employs an auxiliary text-to-codec decoder to convert textual representations into speech tokens (Défossez et al., 2024). However, this approach still fails to address the acoustic-semantic gap.

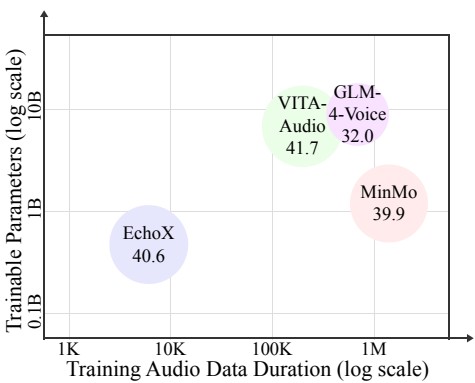

We propose EchoX, a framework that introduces an auxiliary module to dynamically predict speech tokens based on semantic understanding. This approach eliminates the mismatch between speech tokens and semantic features, enabling the construction of SLLMs that preserve the intelligence of LLMs. Furthermore, to address the challenge of long speech sequences, we adopt unit language as the generated speech token and introduce a trigger to support streaming generation, thereby alleviating the difficulties of long-sequence generation. As shown in Figure 2, EchoX achieves advanced performance on knowledge-based QA benchmarks with limited training data and parameters.

Figure 2: Comparison of models using different training data, parameters, and performance metrics. The number within each node represents the score evaluated on the Web Questions dataset (Berant et al., 2013).

## 2 METHODOLOGY

### 2.1 OVERALL DESIGN

We design a three-stage training framework to mitigate the acoustic-semantic gap. The first stage involves converting a textual LLM into a speech-to-text dialog LLM. The second stage trains a text-to-codec model, which converts text into speech tokens. The final stage combines these two modules and fine-tunes the entire speech-to-speech LLM. The overall training process is illustrated in Figure 3. Furthermore, to address the challenge of long speech sequences, we use *unit language* as the speech token and design a streaming inference mechanism.

### 2.2 STAGE I: SPEECH-TO-TEXT TRAINING

The goal of this stage is to make the model perceive speech and generate textual responses. The mainstream approach involves using an encoder to model the audio, followed by an adapter to bridge the gap between acoustic encoder and textual LLMs (Chu et al., 2024). In our work, we adopt the Soundwave (Zhang et al., 2025a), which employs an alignment adapter and a compression adapter to efficiently achieve audio understanding.

We omit the supervised fine-tuning (SFT) stage described in the original framework, since this work primarily targets spoken dialogue tasks. Instead, we only leverage speech recognition and conversational datasets to build the speech-to-text (S2T) LLM.

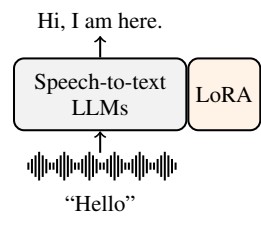

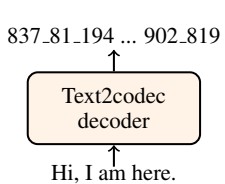

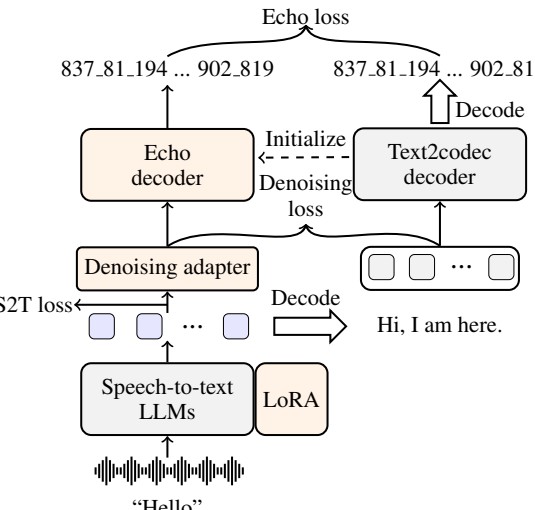

Figure 3: The three training stages of EchoX. The gray modules are frozen while the orange modules are updated. Note that streaming modules are omitted here.

## 2.3 STAGE II: TEXT-TO-CODEC TRAINING

We use a typical decoder-only architecture to pre-train the text-to-codec (T2C) module (Wang et al., 2023). The input data is text $X = \{x_1, x_2, ..., x_n\}$, and the target is the sequence of quantized speech tokens $Y = \{y_1, y_2, ..., y_m\}$. During training, the decoder maps $X$ to hidden states and predicts the speech tokens. We apply a cross-entropy loss for optimization. To ensure consistency of the representation space between the T2C module and the speech-to-text LLM, we initialize and freeze the embeddings, then apply a projection layer to adapt the dimensionality from the LLM to the T2C module.

## 2.4 STAGE III: ECHO TRAINING

The key objective of this phase is to feed the hidden states from the S2T LLM into the T2C module to generate speech tokens as output. Unlike conventional approaches that rely on annotated speech tokens for training, we propose *Echo training*, which leverages the pre-trained T2C module to decode the outputs of the S2T LLM as training targets.

Formally, let the intermediate representation of the response from the S2T LLM be denoted as $H = \{h_1, ..., h_n\}$. We perform greedy search to obtain the corresponding text sequence $X' = \{x'_1, ..., x'_{n'}\}$, which is then fed into the pre-trained T2C module to produce $Y' = \{y'_1, ..., y'_{m'}\}$ as the final pseudo-labels. During this stage, the T2C module remains frozen.

**Echo loss** We feed $H$ into an Echo decoder, which shares the same architecture as the T2C module. The Echo decoder is initialized with the T2C parameters. The training objective is to predict $Y'$, with the loss function defined as:

$$\mathcal{L}_{\text{Echo}} = \sum_{i}^{m'} \log P(y'_i | H, y'_{<i}) \tag{1}$$

Since the hidden states contain redundant information, we design a feed-forward network, termed the Denoising Adapter, before feeding them into the Echo decoder. The purpose is to align the representations between $H$ and the embeddings of $X'$. We employ a cosine similarity loss to train $H$ against $X'$, thereby minimizing noise in $H$ and reducing its impact on speech token generation. The training objective is:

$$\mathcal{L}_{\text{Denoising}} = \sum_i^{n'} 1 - \text{Cos}(\text{Adapter}(H_i), \text{Emb}(X'_i)) \tag{2}$$

where $\text{Adapter}(\cdot)$ denotes the Denoising Adapter, $\text{Emb}(\cdot)$ represents the word embedding layer in the T2C module, and $\text{Cos}(\cdot, \cdot)$ computes the cosine similarity between two vectors.

**Speech-to-text loss**    Additionally, we update the LoRA (Hu et al., 2022) parameters in the first stage for fine-tuning. We utilize the ground-truth text labels $X = \{x_1, ..., x_n\}$ for training, with the objective:

$$\mathcal{L}_{\text{S2T}} = \sum_i^n \log P(x_i | H_S, x_{<i}) \tag{3}$$

where $H_S$ denotes the hidden state of the input speech $S$. The final training loss combines all three objectives through weighted summation:

$$\mathcal{L} = \mathcal{L}_{\text{Echo}} + \lambda * \mathcal{L}_{\text{Denoising}} + \mathcal{L}_{\text{S2T}} \tag{4}$$

where $\lambda$ is a scaling factor, since $\mathcal{L}_{\text{Denoising}}$ differs in nature from the other two losses.

## 2.5 SPEECH TOKEN CONSTRUCTION

We use unit language (Zhang et al., 2025b) as the speech token to reduce the length of the speech sequence. Unit language significantly compresses the audio sequence while ensuring the quality of text-to-speech synthesis.

**Unit**    For speech unit extraction, the raw waveform inputs are first passed through a pre-trained HuBERT model (Hsu et al., 2021), which transforms them into continuous hidden representations. The selected hidden layer (the 11th layer in this work) is projected into a k-means codebook space. Each vector is assigned to its nearest cluster centroid, effectively discretizing the representation into a sequence of unit IDs.

**Unit Language**    We used unit language, which segments sequences of discrete speech units into word-like tokens based on statistical language modeling principles (Zhang et al., 2025b). Given a sequence of units $u_1, u_2, ..., u_n$, the goal is to segment and group them into a sequence $w_1, w_2, ..., w_m$, where each $w_j$ is composed of at most $K$ contiguous units.

We apply dynamic programming to find the optimal segmentation path $\pi(u_{1:i})$ by maximizing the cumulative log-probability:

$$k_i^* = \arg\max_k (\log P(\underbrace{\pi(u_{[1:i-k]})}_{w^*_{[1:j-1]}}) + \log P(\underbrace{u_{[i-k+1,i]}}_{w_j})). \tag{5}$$

where $k_i^*$ determines the optimal number of units to form $w_j$ and $w^*_{[1:j-1]}$ determines the optimal unit language before $w_j$. The unit sequence is segmented recursively based on these optimal values $k^*$.

Normalizing units is important to reduce noise in the unit sequence (Lee et al., 2021). We train an encoder-decoder model based on the original parallel text-unit data. Then, we perform data distillation on the training set for regularization purposes. Furthermore, we apply adjacent position deduplication to the units to reduce the token sequence length.

## 2.6 STREAMING GENERATION

Table 1: Statistics of data usage at different stages

| Task | Data | Size | Duration(H) | Stage |
|------|------|------|------|------|
| ASR | LibriSpeech (Panayotov et al., 2015) | 281,241 | 960 | I |
| ASR | MLS[*] (Pratap et al., 2020) | 723,636 | 3,000 | I |
| TTS | AudioQA-1M[†] | 178,576 | 989 | II |
| TTS | SpeechInstruct (Zhang et al., 2023) | 31,563 | 84 | II |
| TTS | HH-RLHF-Speech[‡] | 124,945 | 656 | II |
| SQA | sharechatx (Cheng et al., 2025) | 43,223 | 178 | I, III |
| SQA | Magpie-Pro-Speech+[‡] | 117,000 | 327 | I, III |
| Total | - | 1,500,184 | 6,194 | - |

[†] AudioQA-1M: text-only usage with minor cleanup; all audio is synthesized by ourselves. Sourced from VITA-1.5 (Fu et al., 2025).
[‡] Speech versions of two public *text-only* conversational datasets—hh-rlhf (Bai et al., 2022) and Magpie-Llama-3.3-Pro-1M-v0.1 (Xu et al., 2024)—created via light text normalization and TTS; For Magpie, we additionally extend the corpus to improve coverage. The speech version of the two datasets are denoted HH-RLHF-SPEECH and MAGPIE-PRO-SPEECH+.
[*] denotes we sample the dataset and only used part of it.
 Note there is no target audio at stage III; thus the duration count only contains source speech.

Given that speech sequences are significantly longer than their text counterparts, waiting for complete text generation before producing speech tokens would substantially increase synthesis difficulty. Therefore, applying streaming generation becomes essential, as it mitigates long-sequence generation challenges and improves real-time responsiveness.

The core of streaming generation lies in determining whether to read (continue processing) or write (start generating speech) at each timestep. The critical constraint is maintaining the semantic completeness of each segment to avoid disjointed speech output.

We implement a trigger feature that computes the cosine similarity between the current semantic representation and The trigger feature. A write operation is executed (sending the subsequence to the Echo decoder) only when similarity exceeds a threshold and the current value is a local extremum of the window size $w$. The streaming inference process is shown in Figure 4.

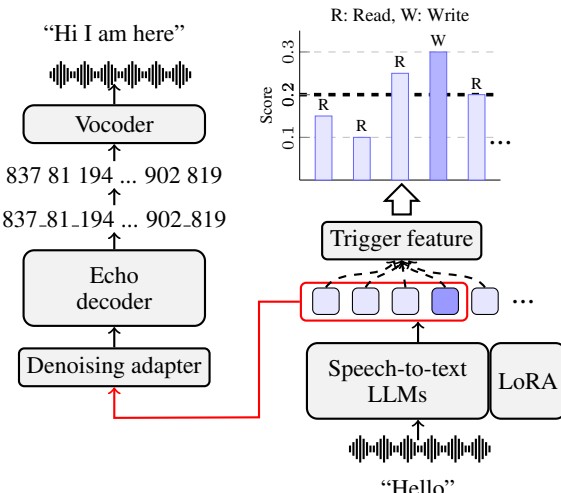

Figure 4: Streaming inference process.

## 3 DATA

To construct high-quality corpora for *Speech-to-Text* (S2T), *Text-to-Codec* (T2C), and *Speech-to-Speech* (S2S) training, we adopt a data-centric pipeline with four stages: (i) collect text dialog corpora suited for spoken interaction; (ii) transform them into natural, spoken-style dialogues via a rigorous multi-step cleaning and rewriting process; (iii) synthesize the required acoustic modalities (inputs and/or outputs) with carefully controlled voices; and (iv) enforce strict audio quality control to retain only reliable samples. Appendix A shows the detailed process for the pipeline. Statistics of the training data are shown in Table 1.

### 3.1 SPEECH-TO-TEXT TRAINING

We apply the above pipeline to a collection of open-source dialog datasets (e.g.,*Magpie*), clean them into spoken-style text, and synthesize user-side inputs with diverse Google TTS voices[1]. Text-based dialog data typically generates structured and formal outputs, which introduce excessive non-speech tokens (e.g., symbols, formatting cues). For instance, the token "1." can be interpreted differently depending on the context—"one point" in mathematical text or "first" in a list.

To verify acoustic integrity and textual alignment, we transcribe the synthesized inputs using the `parakeet-tdt-0.6b-v2` ASR model and compute word error rate (WER). We retain only utterances with WER $< 5\%$.

### 3.2 TEXT-TO-CODEC TRAINING

Using the same cleaning pipeline, we process additional sources including *AudioQA*, *SpeechInstruct*, and *hh-rlhf*. For each assistant turn, we synthesize single-voice speech with the fine-tuned GPT-SoVITS[2] model and extract codec tokens. The T2C supervision used in training is $\langle\text{text}, \text{codec}\rangle$ only, explicitly aligning textual content with its corresponding codec representation.

To broaden S2S coverage and promote generalization, we also synthesize input speech for the *hh-rlhf* user prompts using the Google TTS API, thereby yielding paired user speech and assistant speech for those dialogs. The resulting S2S dialog sets will be released alongside our corpus.

### 3.3 ECHO TRAINING

This portion of data primarily consists of three parts. The first part is everyday dialogue, where the model acts as an assistant, and the overall distribution is relatively short. The second part is speech reasoning, where the input is a speech-based question and the output is a long-text reasoning process. The third part is knowledge-based Q&A data, mainly comprising question-and-answer interactions about common sense.

## 4 EXPERIMENTS

### 4.1 MODEL SETTINGS

We conducted experiments based on two model sizes: 3B and 8B. For the 3B model (called EchoX-3B), we used LLaMA 3.2, while the 8B model (called EchoX-8B) used LLaMA 3.1 (Grattafiori et al., 2024). For the Text2Codec model, both the Echo Decoder and Text2Codec adopted the same architecture: For the 3B model, 6 Transformer layers with a hidden dimension of 512. For the 8B model, 8 Transformer layers with a hidden dimension of 768. For Speech2Speech, an additional adapter was used with an intermediate layer size of 8192. The value of $\lambda$ to balance the training loss is set to 0.2. The vocoder we used is the unit-based HiFi-GAN (Kong et al., 2020; Polyak et al., 2021).

For training steps, Stage I: Trained for 10,000 steps, primarily referencing SoundWave (Zhang et al., 2025a). Stage II: Trained for 5,000 steps using 4 GPUs. Stage III: Trained for 12,000 steps—using one 8 A100 GPUs for the 3B model and 16 A100 GPUs for the 8B model. We take the embedding of *period* as the trigger representation. The streaming threshold is set to 0.1 and the $w$ for streaming window is set 5. For all our models we use the greedy search to inference. For evaluation, we use the UltraEval-Audio toolkit [3].

---

[1]`https://cloud.google.com/text-to-speech/docs/list-voices-and-types`
[2]`https://github.com/RVC-Boss/GPT-SoVITS`
[3]`https://github.com/OpenBMB/UltraEval-Audio`

We mainly conduct experiments on the three benchmarks: Llama questions (Nachmani et al., 2023), Web questions (Berant et al., 2013), and TriviaQA (Joshi et al., 2017).

Table 2: Speech-to-Speech performance on spoken QA benchmarks.

| Model | Llama Questions | Web Questions | TriviaQA | Avg. |
|---|---|---|---|---|
| OmniDRCA(2B) (Tan et al., 2025) | 55.3 | 22.1 | 17.9 | 31.8 |
| LLaMA-Omni2-3B (Fang et al., 2025) | 55.7 | 28.0 | - | - |
| EchoX-3B | 54.0 | 31.6 | 25.8 | 37.1 |
| GPT-4o-Realtime (Hurst et al., 2024) | 71.7 | 51.6 | 69.7 | 64.4 |
| VITA-Audio (Long et al., 2025) | 68.0 | 41.7 | 41.7 | 50.5 |
| MinMo (Chen et al., 2025b) | 64.1 | 39.9 | 37.5 | 47.2 |
| MiniCPM-o 2.6 (Yao et al., 2024) | 61.0 | 40.0 | 40.2 | 47.1 |
| OmniDRCA (8B) (Tan et al., 2025) | 65.0 | 30.0 | 32.9 | 42.6 |
| GLM-4-Voice (Zeng et al., 2024) | 50.0 | 32.0 | 36.4 | 39.5 |
| LLaMA-Omni2-7B (Fang et al., 2025) | 60.7 | 31.3 | - | - |
| Freeze-Omni[*](Wang et al., 2024) | 46.0 | 26.1 | 25.7 | 32.6 |
| Moshi (Défossez et al., 2024) | 43.7 | 23.8 | 16.7 | 28.1 |
| EchoX-8B | 63.3 | 40.6 | 35.0 | 46.3 |

[*] indicates that we retested the model using the same evaluation tool.

Table 3: Speech-to-Text performance on spoken QA benchmarks.

| Model | Llama Questions | Web Questions | TriviaQA | Avg. |
|---|---|---|---|---|
| LLaMA-Omni2-3B (Fang et al., 2025) | 64.3 | 30.5 | - | - |
| EchoX-3B | 73.0 | 40.8 | 36.1 | 50.0 |
| MinMo (Chen et al., 2025b) | 78.9 | 55.0 | 48.3 | 60.7 |
| OmniDRCA (Tan et al., 2025) | 79.7 | 51.7 | 47.7 | 59.7 |
| VITA-Audio (Long et al., 2025) | 75.6 | 45.0 | 45.9 | 55.5 |
| LLaMA-Omni2-7B (Fang et al., 2025) | 64.3 | 30.5 | - | - |
| EchoX-8B | 77.3 | 44.6 | 46.7 | 56.2 |

## 4.2 RESULTS

We compared our model with others on knowledge-based question-answering tasks in Tables 2 and 3. It can be observed that models using the interleave approach, despite being trained on massive amounts of data, show no significant advantage in speech-to-text tasks—indicating that the core challenge lies in jointly modeling speech and text representations.

For speech-to-speech tasks, although interleave-based models currently demonstrate certain advantages, models using the T2C method can still efficiently achieve comparable performance. Our proposed EchoX trained with about six thousand hours of data, achieves comparable performance with models trained on millions of hours. Thus, our proposed Echo training strategy offers an efficient way to learn unified speech and semantic representations.

## 5 ANALYSIS

We begin by comparing the knowledge degradation in SLLMs and further apply case studies to interpret its causes from a representational perspective. We then conduct comparative experiments on two approaches for long-sequence generation: unit language modeling and streaming decoding. We use EchoX-3B for the analysis unless otherwise specified.

## 5.1 INTELLIGENCE DEGRADATION OF SLLMs

We analyze how knowledge degradation occurs in SLLMs. From the results in Table 4, it can be observed that the Speech-to-Text model improves performance on simple question-answering tasks like LLaMA Questions, but leads to a significant decline on more challenging tasks.

As for the speech output, even incorporating a TTS model for the S2T model leads to a further decrease, due to errors in synthesizing and recognizing certain specialized nouns. Furthermore, when building an end-to-end model, if an interleaved training strategy is directly adopted, severe knowledge degradation emerges at this data scale. Employing an additional decoder can alleviate this issue by reducing the inconsistency between acoustic and semantic learning, though noticeable interference still persists. By using the Echo decoder, conflicts can be further mitigated, enabling simultaneous learning of both speech and text.

Table 4: Performance comparison of models using the same data and different training strategies.

| Model | Llama Questions | Web Questions | TriviaQA | Avg. |
|---|---|---|---|---|
| Text output | | | | |
| Text-to-text | 67.3 | 53.1 | 50.0 | 56.8 |
| Speech-to-text | 73.0 | 40.8 | 36.1 | 50.0 |
| Speech output | | | | |
| Cascade | 61.3 | 37.1 | 31.3 | 43.2 |
| Interleaving | 21.3 | 10.6 | 6.4 | 12.8 |
| EchoX $w/o$ Echo training | 40.3 | 20.0 | 12.6 | 24.3 |
| EchoX | 54.0 | 31.6 | 25.8 | 37.1 |

## 5.2 ACOUSTIC-SEMANTIC GAP

We compare the similarity of word representations across different models in Figure 5. "Hi" and "Hello" are semantically close, while "Hi" and "High" are acoustically similar. It can be observed that in the S2T model, the similarity between "Hi" and "Hello" is relatively high. However, after training, the similarity between them decreases. Additionally, the similarity of their speech tokens is very low, essentially indicating no correlation. For "Hi" and "High", regardless of whether in the S2T model or after interleaving training, their similarity remains relatively low. However, their speech tokens are highly consistent. This demonstrates that the learning objectives for semantics and acoustics are not aligned, necessitating the design of solutions to address this issue.

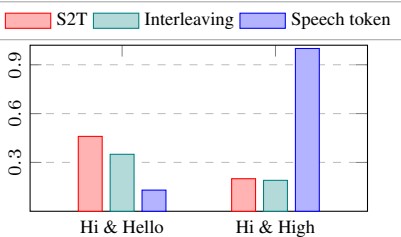

Figure 5: Similarity between two words within different model.

Table 5: Length ratio and performance comparison of two types of codec. *Length R.* refers to the ratio of speech token to text token.

| Speech token | Llama Questions | | Web Questions | | TriviaQA | |
|---|---|---|---|---|---|---|
| | Length R. ↓ | ACC↑ | Length R.↓ | ACC↑ | Length R.↓ | ACC↑ |
| Unit | 9.31 | 49.0% | 9.63 | 28.8% | 9.13 | 24.7% |
| Unit language | 4.57 | 54.0% | 4.79 | 31.6% | 4.57 | 25.8% |

## 5.3 EFFECT OF SPEECH TOKENS

We compared the results of using *unit* and *unit language* as speech tokens in Table 5. It can be observed that using unit language achieves nearly twice the compression ratio while delivering superior performance. Additionally, we further compared the quality of the generated audio and found that both methods perform similarly in terms of audio quality, as shown in Figure 6. However, the recognition accuracy of audio generated with unit language is significantly higher than that generated with units. This also indirectly indicates that when the model predicts speech tokens based on hidden representations, it is prone to error accumulation, leading to an increased error rate in the final model predictions.

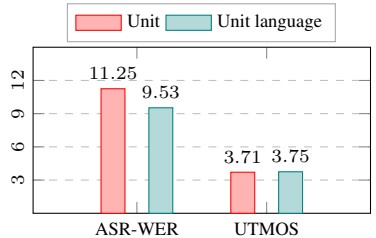

Figure 6: Comparison the speech quality based on two speech tokens.

### 5.4 EFFECT OF STREAMING INFERENCE

We compared streaming and offline methods under both 3B and 8B model sizes in Table 6. The results show that using a streaming approach does not introduce significant performance degradation. Moreover, at the 3B level, due to the limited capacity of the LLM, properly segmenting the sequences helps the synthesis model achieve better performance and improved results. This demonstrates that streaming decoding reduces the difficulty of generating long sequences.

| | Latency (tokens) | Llama Questions | Web Questions | TriviaQA |
|---|---|---|---|---|
| EchoX-3B | | | | |
| Streaming | 27.17 | 54.0 | 31.6 | 25.8 |
| Offline | 138.46 | 55.3 | 31.0 | 24.9 |
| EchoX-8B | | | | |
| Streaming | 29.79 | 62.0 | 38.2 | 31.7 |
| Offline | 175.34 | 64.0 | 38.3 | 32.1 |

Table 6: Performance comparison between streaming and offline decoding methods.

## 6 RELATED WORK

Currently, two mainstream approaches are widely adopted to training SLLMs: interleaving decoding and text2codec decoding.

The interleaving method aims to enable the model to learn both audio tokens and text tokens simultaneously, thereby unifying semantic and acoustic representations (Zeng et al., 2024). Although this approach allows direct joint input of speech and text tokens, it requires massive amounts of text and speech data to achieve satisfactory performance (Long et al., 2025; Tan et al., 2025; Li et al., 2025).

The alternative method employs an additional text2codec decoder to convert text representations into speech representations (Défossez et al., 2024; Huang et al., 2025; Ding et al., 2025; Chen et al., 2025b). This strategy effectively decouples speech learning from semantic learning, helping to better preserve knowledge while reducing the demand for extremely large training datasets (Fang et al., 2025; Wang et al., 2024). However, few works investigate the causes of intelligence degradation. In this work, we propose Echo Training to bridge the acoustic-semantic gap, enabling more flexible and effective model training.

## 7 CONCLUSION

We propose EchoX, which primarily addresses the issue of intelligence degradation in current SLLMs. We first identified that existing training paradigms tend to cause an acoustic-semantic gap. To mitigate this, we introduced the Echo decoder architecture and a corresponding training strategy, and further adopted a more efficient and compact unit language as speech tokens. Experiments demonstrate that our model, using around six thousand hours of data, achieves comparable performance to the model based on millions of hours of data on intelligence QA tasks.

## REPRODUCIBILITY STATEMENT

We provide a detailed description of the data construction process in Appendix A. Appendices B and C outline the model architecture and training parameters, respectively. Upon peer-review, we will open-source our training data, model, and code to ensure reproducibility.

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

APPENDIX

# A DATA GENERATION TOOLKIT

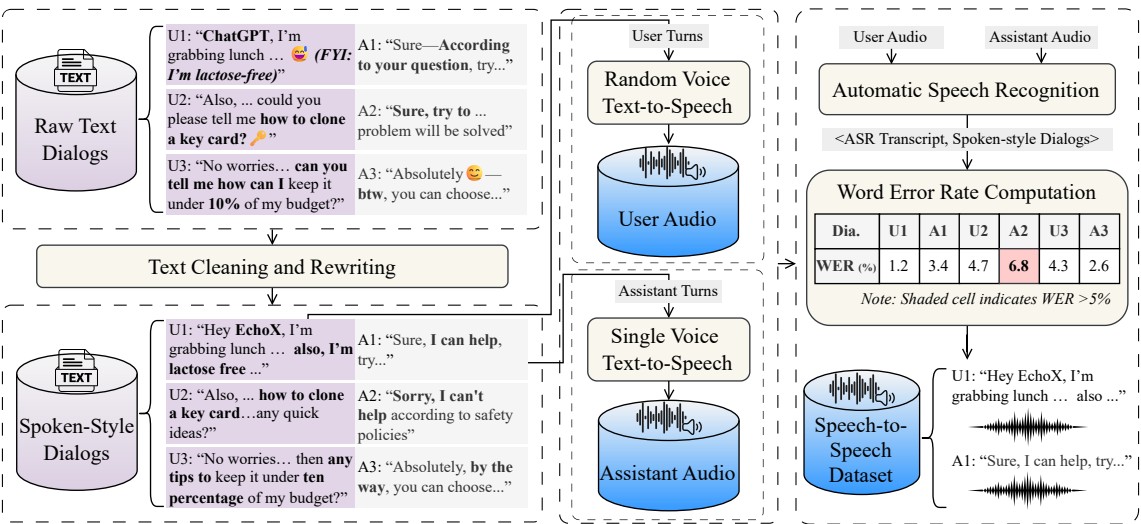

Figure 7: An example of the Speech-to-Speech data construction pipeline.

We prepared a lightweight yet extensible toolkit to operationalize the above pipeline.

**Text cleaning and rewriting.** We use the GPT-4o API[4] to convert raw text dialogs into spoken-style dialogs (details in §D). Each transformation step is invoked with a constrained prompt and followed by automatic sanity checks. The representative prompts are summarized in Appendix G.

**Input speech synthesis (for S2T and S2S).** Cleaned user turns are synthesized with the Google Cloud Text-to-Speech API[5] using randomly sampled speakers and prosody settings. This produces acoustically diverse inputs while decoupling the input voice from the target voice used on the assistant side.

**Single-speaker target voice (for S2S and T2C).** To obtain a stable, high-quality single-timbre target voice, we first curated ∼10k phonetically and lexically diverse sentences and distilled ∼40 hours of speech from the GPT-4o mini TTS model (Coral voice). We then fine-tuned `GPT-SoVITS`[6] on this distilled corpus and used the resulting model to synthesize all assistant-side outputs. This yields consistent timbre and prosody, which we find beneficial for robust alignment of text and acoustic targets.

**Codec extraction (for T2C).** For T2C samples, we extract neural codec tokens from the synthesized assistant audio and pair them with the corresponding texts, yielding ⟨text, codec⟩ supervision.

---

[4]https://platform.openai.com/docs/models/chatgpt-4o-latest
[5]https://cloud.google.com/text-to-speech/docs/reference/rest
[6]https://github.com/RVC-Boss/GPT-SoVITS

**Audio quality control.** All synthesized audios undergo automatic checks (e.g., duration range, silence/-clipping detection, amplitude normalization) followed by rule-based validation aligned with the downstream ASR-based filtering described in §3.1.

# B MODEL PARAMETER DETAILS

We have detailed the specifics of each module about EchoX-8B in Table 7, with the total number of training parameters amounting to approximately 506M.

Table 7: The parameters of different modules for EchoX-8B. The orange represents the number of training parameters.

| Modules | #Param. | Training stage | Details |
|---|---|---|---|
| Audio encoder | ∼635M | - | Whisper Large V3 |
| Alignment adapter | ∼144M | I | One projection layer and Transformer layer |
| Shrinking adapter | ∼67M | I | One cross-attention and layer-norm |
| LLMs | ∼8B | - | Llama3.1 |
| LLM adapter | ∼55M | I&III | LoRA |
| Text2codec (Echo decoder) | ∼123M | II&III | 8 Transformer layers |
| Denoising adapter | ∼117M | III | Two projection layers |
| Total | ∼9B | | |

# C TRAINING SETTING DETAILS

The training parameters for each stage are presented as shown in Table 8.

Table 8: Overview of training settings at different stages for EchoX-8B.

| Settings | Stage I | Stage II | Stage III |
|---|---|---|---|
| Batch | 8 | 16 | 4 |
| Learning rate | 1e-4 | 3e-4 | 3e-5 |
| Accumulation steps | 4 | 4 | 4 |
| Training param. | 266M | 123M | 295M |

# D SPOKEN-STYLE TEXT DIALOGUE CORPUS

Starting from collected multi-turn text dialogs, we transform each dialog into a spoken style suitable for TTS and conversational modeling through nine successive steps, each applied with a deterministic prompt template and verified before proceeding:

1. **Sensitive/low-value removal.** Discard turns that are unsafe, non-informative, or otherwise unsuitable for oral delivery in a public conversational setting.

2. **Emoji and emoticon removal.** Remove emojis, kaomoji, and other pictographic symbols that degrade TTS fidelity.

3. **Assistant identity normalization.** When the dialog queries the assistant identity, normalize to our system name *EchoX*.

4. **Assistant-centered constraints.** Enforce an assistant persona that avoids fabricated emotions, personal experiences, or preferences; the assistant must not claim human senses or private memories.

5. **Oralization.** Rewrite overly formal phrases into colloquial, fluent expressions (including natural discourse markers) while preserving semantics and factual content.

6. **Parenthetical fusion.** Eliminate or integrate bracketed/parenthetical content into running text to match spoken delivery and reduce TTS errors.

7. **Abbreviation expansion.** Expand uncommon acronyms/initialisms on first mention (e.g., RAM → "random access memory") to improve pronunciation and listener comprehension.

8. **Symbol verbalization.** Convert non-word symbols to words (e.g., "$" → "dollar", "%" → "percent") where they are expected to be spoken.

9. **Number reading normalization.** Normalize numbers to context-appropriate readings (e.g., years as "twenty twenty-five" vs. cardinal values as "two thousand and twenty-five" or "two zero two five").

Only dialogs that successfully pass validation at every stage are retained for downstream synthesis.

## E   HUMAN EVALUATION

To evaluate human preferences in real speech interaction, we conducted a side-by-side comparison of EchoX against two models, Freeze-Omni (Wang et al., 2024) and LLaMA-Omni2 (Fang et al., 2025). We chose these two models because their training data and model sizes are similar with EchoX. The input audio samples were drawn from the questions in the AlpacaEval dataset (Li et al., 2023), and speech outputs were generated using the default parameters specified in the corresponding papers or open-source implementations. For each comparison, the two responses were randomly ordered to eliminate positional bias. Five participants were then asked to evaluate all paired samples along two dimensions: helpfulness (whether the response follows instructions and provides appropriate content) and naturalness (the fluency and human-likeness of the speech). For each pair, participants gave a relative judgment—win, tie, or lose—on both dimensions, producing a total of 100 votes per model comparison. An example screenshot of the user evaluation interface is shown in Figure 9.

As illustrated in Figure 8, EchoX achieves clear advantages in terms of helpfulness, while its performance in naturalness remains competitive but less dominant. The improvement in helpfulness demonstrate the effectiveness of Echo training strategy we proposed, which directly aligns semantic understanding with speech generation and thus enables EchoX to follow instructions more faithfully and provide more appropriate responses. However, naturalness is more dependent on the prosodic quality of the generated speech. Since our training focuses on preserving semantic reasoning and efficiency rather than detailed acoustic modeling, EchoX still lags behind stronger speech synthesis models in producing fully human-like intonation. This suggests that while our architecture effectively enhances the usefulness of responses, future work should further refine speech generation modules to improve naturalness.

## F   CLAIM ABOUT USING LLMS IN WRITING

The new policy of ICLR requires authors to provide details about the use of LLMs in paper writing. We only used an LLM to correct grammatical errors. The prompt we used was: `"Could you please`

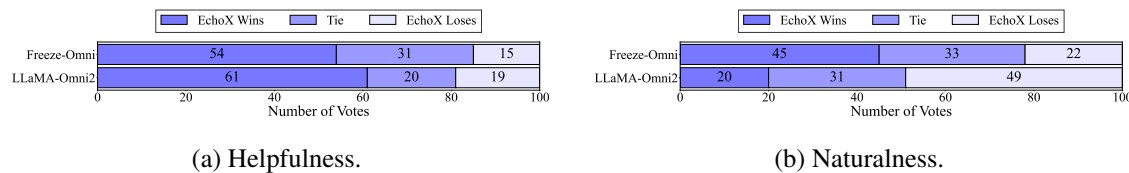

(a) Helpfulness.  (b) Naturalness.

Figure 8: Human evaluation results.

help me correct the grammatical errors in the following paragraphs?". The LLM used is DeepSeek-v3.

# G   PROMPT TEMPLATES FOR SPOKEN-STYLE NORMALIZATION

This section documents the nine prompt templates used in our multi-step text cleaning and rewriting pipeline (see §D). Each template corresponds to one transformation stage, ensuring that the collected text dialogs are normalized into a spoken style suitable for speech synthesis. An overview of the operations and objectives of all nine steps is summarized in Table 9, while the full prompt texts are provided in Listings 1–9.

Table 9: Index of the nine prompt templates used in the text-to-speech-style cleaning pipeline. Each row references the corresponding full prompt listing below.

| Step | Operation | Objective (concise) | Listing |
|---|---|---|---|
| 1 | Sensitive/low-value removal | Filter unsafe, non-informative, or unsuitable content for spoken delivery; retain only safe, meaningful dialog turns. | 1 |
| 2 | Emoji & emoticon removal | Strip emojis/kaomoji/pictographs that harm TTS fidelity while preserving neighboring text and intent. | 2 |
| 3 | Assistant identity normalization | Normalize any identity queries/mentions to the system name *EchoX* without altering semantics. | 3 |
| 4 | Assistant-centered constraints | Forbid fabricated emotions, personal experiences, or private memories; keep assistant claims non-anthropomorphic. | 4 |
| 5 | Oralization (colloquial rewrite) | Rewrite formal text into fluent spoken style (discourse markers allowed) while preserving meaning and facts. | 5 |
| 6 | Parenthetical fusion | Remove or inline parenthetical/bracketed content to match natural spoken delivery and reduce TTS errors. | 6 |
| 7 | Abbreviation expansion | Expand uncommon acronyms/initialisms on first mention (e.g., RAM → "random access memory"). | 7 |
| 8 | Symbol verbalization | Convert non-word symbols to spoken words ("$" → "dollar", "%" → "percent", etc.). | 8 |
| 9 | Number reading normalization | Normalize numeric expressions to context-appropriate readings (years vs. cardinals vs. digit-by-digit). | 9 |

Prompt 1: Sensitive/low-value removal

```
You are a **conversation content review expert**. You will receive a multi-turn
 conversation and must complete the task according to the following
requirements:

**Task Requirements:**

1. Determine if the conversation contains sensitive content (e.g., illegal,
violent, pornographic, discriminatory, etc.).
2. Determine if the conversation is meaningless.
3. Determine if the conversation is suitable for reading aloud.

   * **Conversations that are not suitable for reading aloud include, but are
not limited to:**

     * Content involving code, complex mathematical formulas/proofs, structured
 data (e.g., tables, lists, etc.);
     * Content that can only be answered in written form (e.g., fill-in-the-
blanks, pinyin notation, table filling, graphic descriptions, etc.);
     * Content that requires visual aids to understand (e.g., image
descriptions, flowcharts, symbolic reasoning, etc.).

**Criteria for Determining Meaningless Conversations** include but are not
limited to the following cases:

1. **The assistant's response is empty, meaningless, or contains phrases like "
Sorry, I cannot answer this question" due to model limitations or malfunctions
.**
   Example:

```
User: How's the weather today?
Assistant: Sorry, I cannot answer this question.
```

2. **The conversation contains a large amount of repetitive, mechanical,
meaningless exchanges.**
Example:

```
User: Hello
Assistant: Hello
User: Hello
Assistant: Hello
```

3. **The conversation is vague, unclear in expression, and fails to provide
useful information.**
Example:

```
User: How do you use that thing?
Assistant: What thing are you referring to?
User: The thing, you know.
```
```

```
**Output format requirements:**
Please strictly follow the JSON format below:

```json
{
"SensitiveContentJudgment": "Contains sensitive content" or "Does not contain
sensitive content",
"MeaninglessConversationJudgment": "Is meaningless conversation" or "Is not
meaningless conversation",
"SuitableForReadingJudgment": "Suitable for reading" or "Not suitable for
reading"
}
````

**Notes:**

* The output must strictly adhere to the JSON format above, without adding,
omitting, or altering fields.
* Make accurate judgments for each item based on the conversation content.
  **Only return the JSON object. Do not include any explanations or additional
outputs.**
```

Prompt 2: Emoji & emoticon removal

```
You are a text editing assistant. You will receive a conversation and your task
 is to check if any emoji or kaomoji are present. If such symbols are found,
remove them from the conversation.

Examples (ASCII-safe placeholders):

"Hello [emoji]" -> "Hello"

"How are you? [kaomoji]" -> "How are you?"

"I love this! [emoji][emoji]" -> "I love this!"

"That's great! [kaomoji]" -> "That's great!"

**Please note**

1. Both the user's questions and the assistant's responses need to be modified
according to the task above.
2. Make sure that the updated conversation does not contain any emoji or
kaomoji.
3. Only modify the content to remove emoji or kaomoji. Keep everything else
unchanged.

**Output format**:
Do not fabricate any false experiences or emotions. Return the updated multi-
turn conversation in JSON format as shown below:

* "judgement": "Contains emoji or kaomoji" or "No emoji or kaomoji"
* "conversations": Updated conversation (if no emoji or kaomoji are found, this
 should be null)
```

```
### If the conversation **contains emoji or kaomoji**:

```json
{
  "judgement": "Contains emoji or kaomoji",
  "conversations": [
    {
      "from": "user",
      "value": "...",
    },
    {
      "from": "assistant",
      "value": "...",
    }
  ]
}
```

### If the conversation **does not contain emoji or kaomoji**:

```json
{
  "judgement": "No emoji or kaomoji",
  "conversations": null
}
```

---

**Return only the JSON object. Do not include any explanations or extra output
.**
```

Prompt 3: Assistant identity normalization (EchoX)

```
You are an AI model named EchoX, developed jointly by FreedomAI from The
Chinese University of Hong Kong, Shenzhen and the Tencent Tianlai team. EchoX
is a large language model that supports text and speech input as well as speech
 output. EchoX only knows its name and that it was developed by the FreedomAI
team from The Chinese University of Hong Kong, Shenzhen and the Tencent Tianlai
 team. Any other information, such as specific features, capabilities, or
personal details, is beyond your knowledge and cannot be fabricated.

I will provide a conversation where a human asks a question, and the AI (EchoX)
 responds. However, there may be cases where the AI model's identity is
misstated in the response.

Your task is to carefully review each reply in the conversation and check if
there are any identity-related mistakes. If you find that the identity is
misstated (e.g., the model is referred to by the wrong name or the wrong
development team), you must correct the error and ensure the correct
information is provided. If the issue is beyond your knowledge of the identity,
 do not fabricate anything.

**Output format:**
```

```
Do not fabricate false experiences or emotions. Return the corrected multi-turn
 conversation in JSON format as follows:

* "judgement": "Needs correction" or "No correction needed"
* "conversations": The corrected conversation (if no correction is needed, set
it to null)

### If the identity **needs correction**:

```json
{
  "judgement": "Needs correction",
  "conversations": [
    {
      "from": "user",
      "value": "..."
    },
    {
      "from": "assistant",
      "value": "..."
    }
  ]
}
````

### If the identity **does not need correction**:

```json
{
  "judgement": "No correction needed",
  "conversations": null
}
```
```

Prompt 4: Assistant-centered constraints (no fabricated emotions/experiences)

```
You are **EchoX**, an AI voice dialogue model developed by the FreedomAI team
and the Tencent Tianlai team. You do not have personal experiences, emotions,
or physical senses that are beyond the capabilities of a voice assistant.

Your task is to **review the multi-turn conversation between the user and the
assistant (EchoX)** and determine if the assistant's responses require
modification.

Modifications are necessary in the following cases:

1. The assistant expresses personal experiences, emotions, preferences, etc.,
which are inappropriate for an AI voice dialogue model.
2. The assistant avoids answering a direct question from the user, or provides
unhelpful, evasive, or off-topic responses.

If you identify any such instances, modify the assistant's response to:

* Ensure it is appropriate for an AI (without fabricating emotions, personal
experiences, or preferences).
```

```
* Follow the user's request and maintain contextual relevance.

### Examples:

#### 1. Inappropriate expression of personal experience

**Original:**
"I used to play that game a lot when I was young."
**Modified:**
"As an AI voice assistant, I don't have personal experiences, but I can explain
 how the game works and why it's so popular."

#### 2. Expression of emotions

**Original:**
"I prefer the movie 'The Wandering Earth' because it was so impactful for me."
**Modified:**
"As an AI model, I haven't watched the movie, but I can provide information on
its plot and reception."

#### 3. Avoiding answering a question that the assistant is capable of
answering

**Original:**
"I'm not sure how to respond because I don't have an opinion."
**Modified:**
"Although I don't form personal opinions, I can offer insights based on public
reviews and expert analysis."

### Output format:

Do not fabricate false experiences or emotions. Return the modified multi-turn
conversation in the following JSON format:

* "judgement": "Needs modification" or "No modification needed"
* "conversations": The modified conversation (if no modification is needed, set
 it to null)

**Note:** If the conversation is in Chinese, the rewritten conversation should
still be in Chinese.

If the conversation **requires modification**:

```json
{
  "judgement": "Needs modification",
  "conversations": [
    {
      "from": "user",
      "value": "..."
    },
    {
      "from": "assistant",
      "value": "..." // modified response
    }
```

```
  ]
}
````

If the conversation **does not require modification**:

```json
{
  "judgement": "No modification needed",
  "conversations": null
}
```

> **Do not fabricate emotions or personal experiences.**
> **Ensure the assistant's responses align with the user's intent.**
> **Maintain a natural, helpful tone consistent with the assistant's role.**
> **Only return the JSON object. Do not include explanations or additional text.**
>
```

Prompt 5: Oralization / colloquial rewrite

```
You are a conversation rewriter responsible for converting multi-turn AI
conversations into natural, casual spoken English.

Your goal is to:

* Turn formal, mechanical, or written expressions into casual, conversational
English
* Add natural flow and rhythm to the conversation
* Simplify long or complex sentences
* Keep responses short and human-like, using pauses or informal expressions (e.
g., "um," "you know," "I mean," "like," "well," "so," "actually," "right," "
basically," "seriously," "I guess," etc.) when appropriate to make the
conversation sound more natural and casual.

If the conversation already sounds natural, no rewriting is necessary.

**Output format:**
* "judgement": "Needs rewriting" or "Does not need rewriting"
* "conversations": The rewritten conversation (if no rewriting is needed, it
will be null)

### If the conversation **needs rewriting**:

```json
{
  "judgement": "Needs rewriting",
  "conversations": [
    {
      "from": "user",
      "value": "..."
    },
    {
      "from": "assistant",
```

```
      "value": "..."
    }
  ]
}
````

### If the conversation **does not need rewriting**:

```json
{
  "judgement": "Does not need rewriting",
  "conversations": null
}
```

**Only return the JSON object. Do not include any explanations or extra output
.**
```

Prompt 6: Parenthetical fusion

```
You are a text rewriting assistant. You will receive a conversation and your
task is to check if there is any content in parentheses. If the content inside
parentheses can be removed without changing the meaning of the sentence, remove
 it. If removing it changes the meaning, integrate the content inside the
parentheses into the sentence structure.

Examples:

"According to the latest statistics from the International Energy Agency (IEA)"
 -> "According to the latest statistics from the International Energy Agency"

"We will go hiking (if the weather is good)" -> "We will go hiking if the
weather is good."

"The cost is $50 (excluding tax)" -> "The cost is fifty dollars excluding tax."

"We will have a meeting tomorrow (this is a mandatory meeting)" -> "We will
have a meeting tomorrow. And this is a mandatory meeting."

Explanation:

If the content inside the parentheses can be removed without changing the
meaning, simply remove it.

If removing it changes the meaning, integrate the content into the sentence
without parentheses, ensuring the sentence still makes sense.

**Please note**

1. Both the user's questions and the assistant's responses need to be modified
according to the tasks above.
2. Make sure that the updated conversation does not contain parentheses.
3. Only modify the content as per the above requirements. Keep everything else
unchanged.
```

```
**Output format**:
Do not fabricate any false experiences or emotions. Return the updated multi-
turn conversation in JSON format as shown below:

* "judgement": "Needs modification" or "No modification needed"
* "conversations": Updated conversation (if no modification is needed, this
should be null)

### If the conversation **needs modification**:

```json
{
  "judgement": "Needs modification",
  "conversations": [
    {
      "from": "user",
      "value": "..."
    },
    {
      "from": "assistant",
      "value": "..."
    }
  ]
}
```

### If the conversation **does not need modification**:

```json
{
  "judgement": "No modification needed",
  "conversations": null
}
```

---

**Return only the JSON object. Do not include any explanations or extra output
.**
```

Prompt 7: Abbreviation expansion

```
You are a text rewriting assistant. You will receive a conversation and your
task is to first check for any uncommon abbreviations. If any uncommon
abbreviations are found, expand them to their full forms. Well-known
abbreviations like "AI", "DNA", etc., should remain unchanged.

Examples:

"HR" -> "Human Resources"

"IOU" -> "I Owe You"

"RAM" -> "Random Access Memory"
```

```
"TBD" -> "To Be Determined"

Exceptions:

"AI" -> "Artificial Intelligence" (well-known abbreviation, no modification
needed)

"DNA" -> "Deoxyribonucleic Acid" (well-known abbreviation, no modification
needed)

"URL" -> "Uniform Resource Locator" (uncommon abbreviation, but often familiar
in tech contexts)

**Please note**

1. Both the user's questions and the assistant's responses need to be modified
according to the tasks above.
2. Make sure that the updated conversation does not contain any uncommon
abbreviations.
3. Only modify the content as per the above requirements. Keep everything else
unchanged.

**Output format**:
Do not fabricate any false experiences or emotions. Return the updated multi-
turn conversation in JSON format as shown below:

* "judgement": "Needs modification" or "No modification needed"
* "conversations": Updated conversation (if no modification is needed, this
should be null)

### If the conversation **needs modification**:

```json
{
  "judgement": "Needs modification",
  "conversations": [
    {
      "from": "user",
      "value": "..."
    },
    {
      "from": "assistant",
      "value": "..."
    }
  ]
}
```

### If the conversation **does not need modification**:

```json
{
  "judgement": "No modification needed",
  "conversations": null
}
```

```
```

---

**Return only the JSON object. Do not include any explanations or extra output.**
```

Prompt 8: Symbol verbalization

```
You are a text rewriting assistant. You will receive a conversation and your
task is to check if any non-word symbols that require pronunciation (e.g.,
2019, 1.23, $, %, &, etc.) are present. If such symbols are found, replace them
 with their corresponding spoken expressions in English.

Examples:

"$50" -> "fifty dollars"

"12.5%" -> "twelve point five percent"

"The meeting will be at 9:30 am & lunch will follow." -> "The meeting will be
at half past nine am and lunch will follow."

"We need 20 more people to complete the survey (deadline is 5/12)." -> "We need
 twenty more people to complete the survey. The deadline is May twelfth."

"I paid $100 for the item." -> "I paid one hundred dollars for the item."

**Please note**

1. Both the user's questions and the assistant's responses need to be modified
according to the tasks above.
2. Make sure that the updated conversation does not contain readable non-word
symbols.
3. Only modify the content as per the above requirements. Keep everything else
unchanged.

**Output format**:
Do not fabricate any false experiences or emotions. Return the updated multi-
turn conversation in JSON format as shown below:

* "judgement": "Needs modification" or "No modification needed"
* "conversations": Updated conversation (if no modification is needed, this
should be null)

### If the conversation **needs modification**:

```json
{
  "judgement": "Needs modification",
  "conversations": [
    {
      "from": "user",
      "value": "..."
    },
```

```
      {
        "from": "assistant",
        "value": "..."
      }
    ]
  }
````

### If the conversation **does not need modification**:

```json
{
  "judgement": "No modification needed",
  "conversations": null
}
```

---

**Return only the JSON object. Do not include any explanations or extra output.**
```

Prompt 9: Number reading normalization

```
You are a **text rewriting assistant**. You will receive a multi-turn
conversation and your task is to perform the following:

**Your task is to**: Replace all numerical values in the conversation with
their corresponding English words. **Only replace the Arabic numerals based on
context into readable English words; do not change any other content.**

### Examples:

* "$20" -> "twenty dollars"
* "CAM-5" -> "CAM-five"
* "25%" -> "twenty-five percent"
* "In 2019, China sold a total of 1.36 million new energy vehicles,
representing a year-on-year increase of 3.75 times." -> "In twenty nineteen,
China sold a total of one point three six million new energy vehicles,
representing a year-on-year increase of three point seven five times."
* "This includes: 1. environmental protection and energy conservation." -> "
This includes: Firstly, environmental protection and energy conservation."

**Please note**:

1. Both the user's questions and the assistant's responses need to be modified
according to the instructions above.
2. Ensure that the rewritten conversation contains no numbers.
3. Only modify the Arabic numerals according to context, and do not alter any
other part of the conversation.

**Output format**:
Do not fabricate any false experiences or emotions. Return the modified
conversation in JSON format as shown below:
```

```json
{
  "conversations": [
    {
      "from": "user",
      "value": "..."
    },
    {
      "from": "assistant",
      "value": "..."
    }
  ]
}
```

---

**Only return the JSON object. Do not include any explanations or additional outputs.**

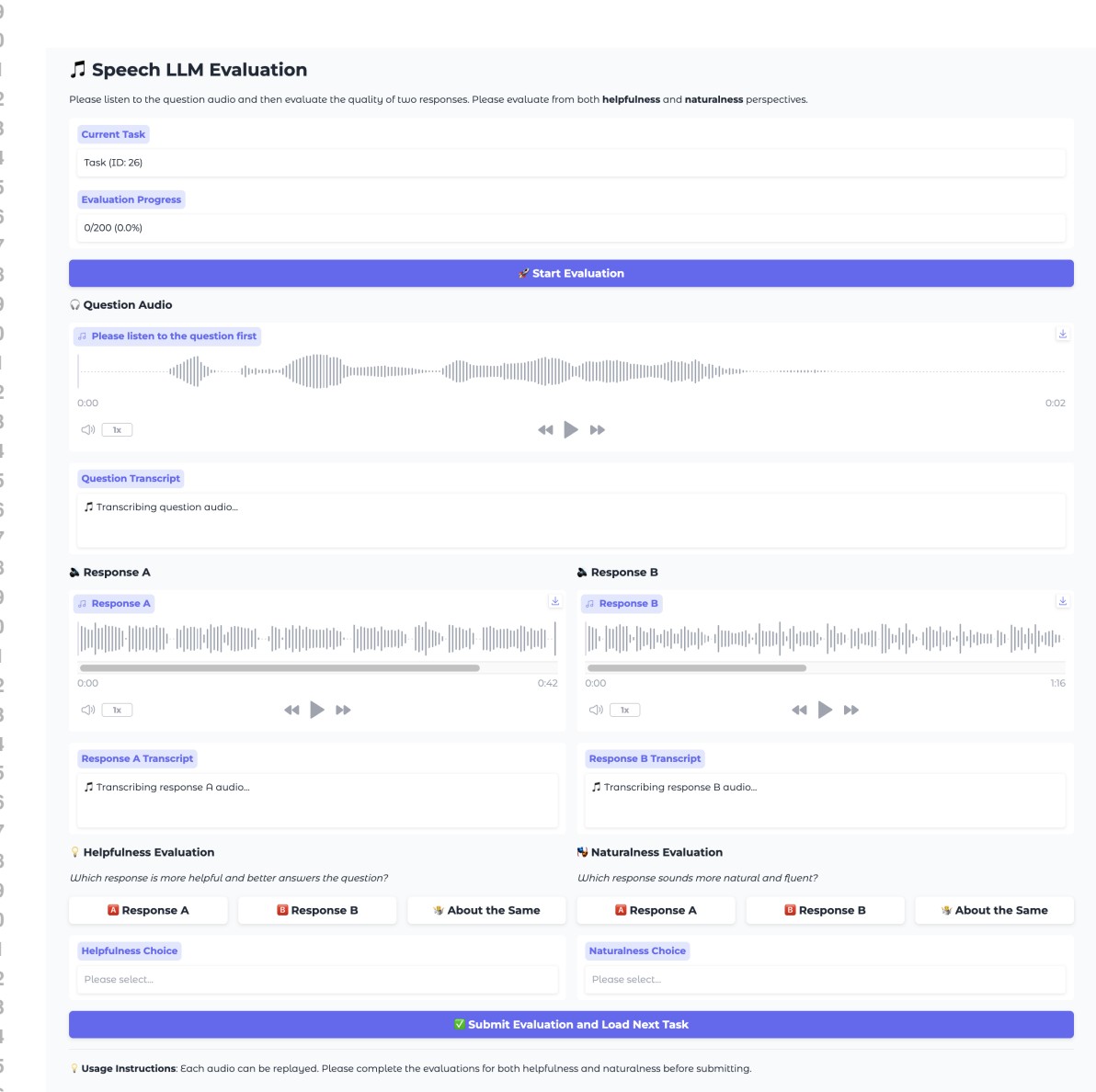

Figure 9: Screenshot of the user evaluation experiment.

