# OpenReview forum: "EchoX: Towards Mitigating Acoustic-Semantic Gap via Echo Training for Speech-to-Speech LLMs"
_ICLR.cc/2026/Conference — Submitted to ICLR 2026_

### Official Review · Reviewer_uHyc · 2025-10-24

**Soundness:** 3
**Presentation:** 3
**Contribution:** 2
**Rating:** 2
**Confidence:** 5

**Summary:**

This paper proposes EchoX, a three-stage pipeline for speech-to-speech LLMs. The key idea is to use a frozen text-to-codec module to generate pseudo speech targets from intermediate semantic representations, with the goal of reducing the acoustic-semantic gap. The model also adopts unit-language tokens and a trigger-based streaming mechanism. Experiments on spoken QA benchmarks show competitive results with relatively modest training data.

**Strengths:**

* Targets a real issue in SLLMs (knowledge degradation).
* Unit-language tokens help reduce sequence length without hurting quality.
* The data pipeline could be useful to the community, and planned release is valuable.

**Weaknesses:**

1. The method feels close to a cascaded S2T + TTS system with an extra alignment step. The novelty and contribution are not very clear.
2. The main claim (mitigating the acoustic–semantic gap) is not convincingly demonstrated. I would expect clearer evidence showing reduced semantic degradation compared to text-only or other S2S approaches. In particular, the paper does not show that the gap between S2T and S2S is actually smaller than in other systems. For example, the reported S2T vs. S2S scores are 77.3 vs. 63.3, while VITA-Audio shows 75.6 vs. 68.0, indicating a narrower gap than the proposed model. Maybe the paper can add WER results and S2T/S2S relative gap like [1] mentioned to show it clearly.
3.  I have concerns about the fairness and stability of the test set. The accuracy on these spoken QA benchmarks is computed via keyword matching, which leads to inconsistent results across papers for the same model (e.g., GLM-4-Voice, LLaMA-Omni2). Some of questions also not suitable for testing, e.g. rely on chemical symbols or other abbreviated forms, which makes the score highly sensitive to ASR and text normalization errors rather than the S2S generation quality itself.

[1] Towards Efficient Speech-Text Jointly Decoding within One Speech Language Model https://arxiv.org/pdf/2506.04518v1

**Questions:**

Overall, while the paper targets an important problem: the intelligence degradation introduced during modality conversion, and the dataset release is a valuable contribution to the community. But I remain skeptical that the proposed method effectively addresses this issue. The evidence presented does not convincingly support the main claim.

Below are several technical questions for the authors:
1. The Echo loss uses speech codec targets obtained by decoding the S2T hidden states into text via greedy search and then passing them through a frozen T2C model. Since the T2C module is pre-trained independently and does not update during Stage III, all speech-token supervision could be precomputed offline by running T2C on ground-truth transcripts. Could the authors clarify what representational benefit the online speech label generation provides? In particular, what gradient differences arise compared to purely offline codec labels?
2. Semantic–Acoustic Gap Quantification. Echo training is motivated by reducing the acoustic–semantic gap in representation space, yet the paper measures only downstream QA accuracy. Could the authors provide quantitative metrics validating that H is closer (e.g., via cosine similarity, probing, clustering entropy) to semantic spaces after Echo training than before? Without such evidence, it is unclear whether Echo training modifies representation geometry rather than simply regularizing the decoder.
3. In an AR setup with NTP loss and causal masking, the model can already leverage corrected previous text and speech tokens during training to predict the token at timestep t. When paired with offline text labels and corresponding TTS-generated codec tokens, the model receives clean supervision at every step. Given this, it is unclear what additional benefit the Echo loss provides beyond standard AR conditioning. Could the authors clarify what unique learning signal Echo loss introduces that is not already captured by the lower-triangular causal mask and NTP supervision?
4. Could the paper add benchmark like Voicebench? It would strengthen the empirical evidence and make the results more convincing.

---

### Official Review · Reviewer_rSNH · 2025-10-29

**Soundness:** 2
**Presentation:** 2
**Contribution:** 1
**Rating:** 2
**Confidence:** 4

**Summary:**

This paper proposes a multi-stage training framework of speech-to-speech large language models (SLLMs) in order to bridge the acoustic-semantic gaps. The proposal is based on the insight that due to acoustic-semantic gaps, current SLLMs have not achieved on-par 'intelligence' between text and speech modality.
The main contribution of the paper is that the proposed framework introduces a module to dynamically predict speech tokens based on semantic input, aiming to bridge the gap between output speech tokens and the semantic features.

**Strengths:**

Originality:
- The author lays out clear motivation of the proposal - assuming acoustic-semantic gap is the 'intelligence degradation' in SLLMs, and turns that insight into architectural solution (Echo training).

- The proposed framework utilizes a pretrained text to codec (T2C) model to generate pseudo  speech tokens from text, which reduces the demand for speech-to-speech data which is relatively scarce.

- The proposed framework introduces a streaming inference mechanism with read/write trigger.

Clarify:
- The multi-stage framework (S2T, T2C, joint S2T with echo training) appears to be straightforward and modular recipe which combines the conventional methods in an organic way. It's easy to follow.

**Weaknesses:**

1. The claim
There should be a rigorous definition of "acoustic-semantic gap". Moreover, the claim of "acoustic-semantic gap" leading to "intelligence degradation" is thin. Figure 1 itself is not a sufficient demonstration of the concept, more evidence should be provided to argue that a)"acoustic-semantic gap" *causes* "intelligence degradation" b) most of (if not all) SLLMs with various choices of speech tokenizers have this  "acoustic-semantic gap" issue. If such evidence exists in other literatures, please quote them.

2. Potential design flaws. There are some design choices in EchoX framwork that are questionable, please find in 'Question' section.

3. Experimental setup: there's lack of detailed ablation study and more general analysis. Please find in 'Question' section.

4. Presentation needs to improve. For instance, in formula (1) and (3) represent log likelihood of the target sequence, however as a loss function to minimize using gradient descent, it's supposed to be instead *negative* log likelihood.

**Questions:**

1. Some speech tokenizers focus on semantic and some focus on acoustic, others try to balance both. Of all these speech tokenizers, do all of them have this "acoustic-semantic gap" issue? Is it more of an issue for Speech LLMs or the choice of speech tokenizers? In another word, how to prove that EchoX training method could universally improve most of the speech LLMs?
2. In section 2.4, why do you do greedy search to get pseudo text labels $X'$ when you have ground truth text? Wouldn't that cause error prorogation?
3. In Table 4: what's EchoX without Echo training? is it essentially a cascaded model? Why is Speech-to-text scores higher than Text-to-text on "Llama Questions"?
4. In Table 4: the details of 'interleaving' should be given, how did authors make sure they are comparable?
5. Lacking ablation study:
- The pre-trained modules such as T2C and S2T LLMs need evaluation.
- the effect for denoiser
6. If speaker variance is out-of-scope for this work, how would authors argue that the proposed framework could generalize and solve 'acoutsic-semantic gap' in real-word applications?

---

### Official Review · Reviewer_bm8h · 2025-10-29

**Soundness:** 2
**Presentation:** 2
**Contribution:** 2
**Rating:** 2
**Confidence:** 2

**Summary:**

Based on the observation that speech LLM training degrades the text capabilities of an LLM, the paper tries to address this issue by a text2codec guidance during training of the speech LLM. The speech codecs are based on grouping of HuBERT units. The paper proposes a multi-stage training strategy given a pre-trained text LLM. First the model is trained for Speech to text, then a text to speech codec (T2C) module is trained, finally the proposed Echo training uses the guidance from the T2C module. The model takes in input speech, passes it through the S2T LLM, outputs of this step are fed into both a text denoising layer and to the T2C module. The training objective then combines the losses of consistency of the text output and the T2C embeddings, T2C outputs with the speech token decoding outputs, and the S2T loss. The model is trained on about 6k hours of data and the experimental evaluations suggest competitive performance on Llama Questions, Web Questions, TriviaQA speech QA tasks. The scores are slightly behind other models in literature (e.g. GPT-4o-Realtime (Hurst et al., 2024), VITA-Audio (Long et al., 2025), MinMo (Chen et al., 2025b) ) in both its 3B and 8B versions. The paper than demonstrates an example where the semantic and acoustic gap is happening (hello - hi - high).

**Strengths:**

Originality
* The way the Echo loss is constructed might be novel. At an intermediate level, the text tokens are mapped to speech tokens which guide the learning of the S2S model. Construction of the unit language is based on the Soundwave paper, the LLM backbone is based on Llama, and the decoder consists of some transformer layers, hence the model architecture is not particularly novel.

Quality
* The example given in Section 5.2. is successful at demonstrating the acoustic-semantic gap problem.

* According to the experiments, with 6k hours of synthetic speech data it is possible to achieve competitive performance as compared to other models requiring much more training data.

Clarity
* There is no major concern around clarity of the writing. However, the flow of the paper could have been improved.

Significance
* The paper is relevant to the speech LLM community which focuses on speech and text domain alignment issues.

* The paper claims that the generated synthetic dataset will be made available, which may become a useful source.

**Weaknesses:**

Originality
- The originality is limited to the design of Echo loss. Architectural components and experimental design have been introduced before.

Quality
- Even though the results are competitive with other models trained on much more data, the results do not provide the latest SOTA. In addition, the three datasets may not necessarily reveal the general performance of the model on datasets with different difficulty levels.

- Evaluation setup is somewhat limited. Especially, for S2S tasks, only the QA performance is presented. It could be informative to also see the speech quality metrics such as MOS scores.

- Since the paper is trying to address the loss of reasoning/intelligence capabilities, it could be good to demonstrate the performance on unseen speech tasks.

Clarity
- The example discussed in Section 5.2. could have been shown in the introduction to better motivate/explain the problem that the model is addressing.

**Questions:**

1. Did the authors measure the speech quality metrics (e.g. MOS) for the S2S model in addition to the content evaluation? How would it compare to a S2T and a TTS cascade?

2. Did the authors check the generalization capabilities of the model to unseen tasks and datasets at different difficulty levels?

3. Do the authors have any comment on the use of synthetic speech data instead of real speech data in model training? Did the authors experiment with any real spoken QA datasets?

4. The simple example provided in Section 5.2. is informative, however, the placement of this analysis to an earlier place in the paper may provide a better motivation for the reader.

5. Table 5 caption introduces the $R$ measurement, which was not discussed in the text before.

---

### Official Review · Reviewer_fK1Q · 2025-11-01

**Soundness:** 2
**Presentation:** 2
**Contribution:** 2
**Rating:** 4
**Confidence:** 3

**Summary:**

This paper proposes EchoX, a three-stage training framework designed to mitigate the acoustic-semantic gap in Speech-to-Speech Large Language Models (SLLMs). Its core technical contribution is "Echo Training," where an Echo decoder, initialized from a pre-trained Text-to-Codec (T2C) model, is trained to generate speech tokens from the hidden states of a Speech-to-Text (S2T) LLM, using dynamically generated pseudo-labels from the same T2C model. The method also employs a denoising adapter and a streaming inference mechanism. The model is evaluated on knowledge-based QA benchmarks and shows competitive performance with only ~6k hours of training data.

**Strengths:**

- The focus on intelligence degradation in SLLMs is  a well motivated problem.
- The paper provides extensive details on the data pipeline and model configurations.

**Weaknesses:**

- The paper fails to evaluate speech generation quality beyond accuracy on QA tasks.
- The paper fails to discuss and contrast its approach with highly relevant work, such as SpeechGPT.  And comparisons to other recent strong baselines like Qwen-Audio are missing.
- The proposed method has several new components (Echo Training, Denoising Adapter). However, there is no rigorous ablation study to isolate the contribution of each.
- While Figure 5 attempts to illustrate the acoustic-semantic gap, the analysis is superficial. The choice of words ("Hi", "Hello", "High") is simplistic and not representative of the complex semantic-acoustic interactions in real dialogue.

**Questions:**

- Why did you not include speech quality metrics (e.g., MOS, WER) to evaluate the generated audio? Without these, the claim of “mitigating the acoustic-semantic gap” is only partially supported.
- How does EchoX quantitatively and qualitatively compare to SpeechGPT, which also uses a three-stage pipeline and unit tokens?
- What is the performance drop if you remove the Denoising Adapter? What if you train the Echo decoder directly on ground-truth speech tokens instead of the T2C-generated pseudo-labels?

---

### Meta-Review · Area_Chair_aDmg · 2025-12-09

**Summary:**

This paper proposes "EchoX," a framework designed to mitigate the "acoustic-semantic gap" in Speech-to-Speech Large Language Models (SLLMs).

### Pros

* The focus on intelligence degradation in SLLMs is a well motivated problem.
* The paper provides extensive details on the data pipeline and model configurations.

### Cons
* Insufficient Evaluation
* Weak Support for Main Claim
* Missing Baselines & Ablations
### AC's evaluation

1. from reviews and rebuttals

This paper receives 4222. All reviewers agree with rejection. The reviews present a clear consensus for rejection. No rebuttals are provided.

2. from AC's reading

I recommend Rejection. The paper's premise is interesting, but the empirical validation falls short of the standard required for ICLR.

**Reviewer Concerns:**

All Concerns are still outstanding.; the author did not provide a rebuttal.

**Reviewer Scores:**

Not applicable; the author did not provide a rebuttal.

---

### Decision · Program_Chairs · 2026-01-26

Reject